# A Query-Parallel Machine Reading Comprehension Framework for Low-resource NER

**Yuhao Zhang**
Alibaba Group
zyh277500@taobao.com

**Yongliang Wang**
Alibaba Group
wangyongliang.wyl@taobao.com

## Abstract

Named entity recognition (NER) is a fundamental task in natural language processing. Recently, NER has been formulated as a machine reading comprehension (MRC) task, in which manually-crafted queries are used to extract entities of different types. However, current MRC-based NER techniques are limited to extracting a single type of entities at a time and are largely geared towards resource-rich settings. This renders them inefficient during the inference phase, while also leaving their potential untapped for utilization in low-resource settings. We suggest a query-parallel MRC-based approach to address these issues, which is capable of extracting multiple entity types concurrently and is applicable to both resource-rich and resource-limited settings. Specifically, we propose a query-parallel encoder which uses a query-segmented attention mechanism to isolate the semantics of queries and model the query-context interaction with a unidirectional flow. This allows for easier generalization to new entity types or transfer to new domains. After obtaining the query and context representations through the encoder, they are fed into a query-conditioned biaffine predictor to extract multiple entities at once. The model is trained with parameter-efficient tuning technique, making it more data-efficient. We conduct extensive experiments and demonstrate that our model performs competitively against strong baseline methods in resource-rich settings, and achieves state-of-the-art results in low-resource settings, including training-from-scratch, in-domain transfer and cross-domain transfer tasks.

## 1 Introduction

Named entity recognition (NER) is a fundamental task in natural language processing, aiming at detecting the text spans that refer to entities. It has been widely used in various downstream tasks, such as entity linking (Martins et al., 2019; Tedeschi et al., 2021), relation extraction (Miwa

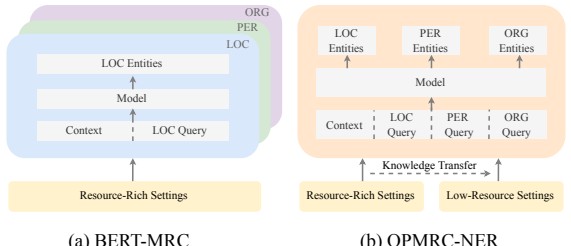

(a) BERT-MRC      (b) QPMRC-NER

Figure 1: Comparison between machine reading comprehension based NER frameworks. (a) The BERT-MRC method extracts one type of entities at a time and is primarily employed in standard resource-rich supervised settings. (b) The proposed QPMRC-NER is capable of recognizing multiple entity types simultaneously and is effective in both resource-rich and resource-limited settings.

and Bansal, 2016; Zhong and Chen, 2021) and information retrieval (Cheng et al., 2021).

Traditionally, NER has been formalized as a sequence labeling task, assigning a single tag class to each token in a sentence (Chiu and Nichols, 2016; Ma and Hovy, 2016; Xia et al., 2019; Devlin et al., 2019; Lin et al., 2020). Recently, several new NER paradigms have been proposed, which conceptualize NER as span classification (Luan et al., 2019; Yu et al., 2020; Li et al., 2021; Shen et al., 2021; Fu et al., 2021a), sequence generation (Straková et al., 2019; Yan et al., 2021; Tan et al., 2021; Paolini et al., 2021; Lu et al., 2022) and constituency parsing (Fu et al., 2021b; Lou et al., 2022) tasks. In spite of achieving promising performance, these approaches require large amounts of well-annotated in-domain data (Lison et al., 2020; Ma et al., 2022). The data annotation usually involves carefully defined guideline and annotators with domain expertise, which could be quite time-consuming and cost-prohibitive. As a result, the development of NER systems is a costly endeavor in real-world scenarios where usually only a small amount of labeled data is available for new domains (Huang et al., 2021).

Recently, some works reformulate NER as a ma-

chine reading comprehension (MRC) task (Li et al., 2020; Mengge et al., 2020). As illustrated in Figure 1(a), the input sentence is regarded as the context, a collection of queries that have been manually crafted to represent various entity types can be viewed as questions, then extracting entities from the context could be solved by employing question answer techniques. Prior MRC-based NER approaches are limited to extracting one type of entities at a time and have largely been applied to the standard supervised setting. In this context, they have exhibited superior performance in comparison to conventional sequence labeling techniques and have demonstrated comparable performance with reduced training data. Nonetheless, the extraction of a singular entity type at a time is not particularly efficient, and their potential for deployment in low-resource settings remains largely unexplored.

To alleviate these issues, this paper presents a query-parallel machine reading comprehension framework for named entity recognition (QPMRC-NER). It is suitable for both resource-rich and resource-limited settings, and is characterized by enhanced efficiency, as illustrated in Figure 1(b). It consists of two key components: the query-parallel encoder and the query-conditioned biaffine predictor. The query-parallel encoder takes the combination of the context sentence and the queries with shared continuous prompt as input, and utilizes a query-segmented attention mechanism to separate the queries from one another and model the query-context interaction with a unidirectional flow, thus facilitating easier generalization to new entity types and domain transfer. After obtaining the contextualized representations from the query-parallel encoder, we feed them to the query-conditioned biaffine predictor to extract entities of multiple types simultaneously. The model is trained with parameter-efficient techniques for data-efficiency (Li and Liang, 2021; Pan et al., 2022). We conduct extensive experiments, and in most cases, our model outperforms the present SOTA methods.

The main contributions of this paper are summarized as follows: (1) We introduce a novel query-parallel machine reading comprehension framework QPMRC-NER that is capable of handling low-resource NER tasks effectively and efficiently. (2) Our MRC-based architecture facilitates simultaneous extraction of entities belonging to multiple categories, resulting in faster inference speed. (3) We conduct extensive evaluations of

QPMRC-NER across a diverse range of NER tasks. Our model exhibits competitive performance in resource-rich settings and achieves state-of-the-art results in low-resource settings, including training-from-scratch, in-domain transfer, and cross-domain transfer tasks.

## 2 Method

This section presents the task formulation in Section 2.1, then introduces the proposed method QPMRC-NER, illustrated in Figure 2. QPMRC-NER consists of two components: the query-parallel encoder and the query-conditioned biaffine predictor, explained in Section 2.2 and Section 2.3 respectively.

### 2.1 Task Formulation

Given an input sentence $S = \{w_1, w_2, ..., w_n\}$ with sequence length $n$, the goal is to extract all entities $L = \{< I_i^{start}, I_i^{end}, T_i^{tag} >\}_{i=0}^m$ from it. Here, $I_i^{start}$ and $I_i^{end}$ indicate the start and end positions of the $i$-th entity span, $T_i^{tag}$ denotes the type of the $i$-th entity which belongs to a finite set of entity types $\mathcal{E}$. Our method utilizes a query $Q$ for each entity type to extract entities from the input sequence. Thus, the task can be formulated as extracting all entities $L$ from $S$ based on queries $\{Q_1, Q_2, ..., Q_{|\mathcal{E}|}\}$.

### 2.2 Query-Parallel Encoder

In order to extract different types of entities simultaneously with the machine reading comprehension framework, we concatenate the input sentence and queries and feed them into a transformer-based encoder, from which we obtain the context word representations and entity type representations that are used for entity prediction.

**Query Generation and Model Inputs** Conventional MRC-based NER methods rely on manually-crafted queries to represent each entity type, which often requires domain expertise and laboriously tuning of query words, rendering it non-reusable for new entity types.

Rather than relying on manually tuning, we construct queries using a shared prompt prefix that is composed of a set of learnable vectors. This approach renders the prompt more suitable for NER task and facilitates generalization when dealing with new entity types. So the query for the $i$-th entity type is $Q_i = \{p_0, p_1, ..., p_m, e_{i,1}, ..., e_{i,n}\}$, where $p_0, p_1, ..., p_m$ represent the shared learnable

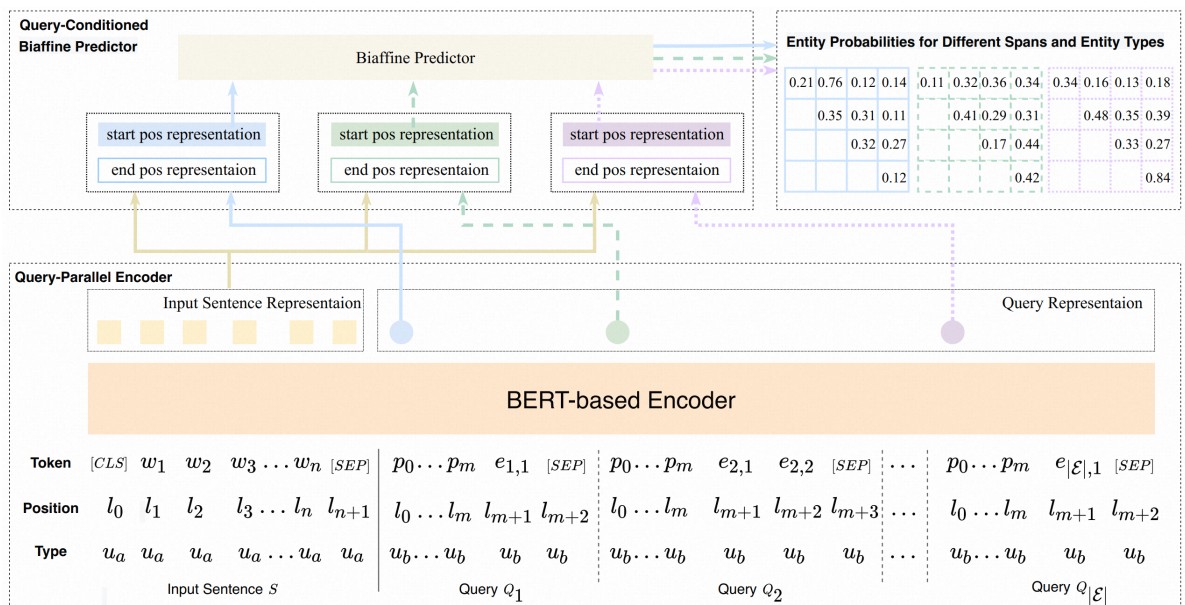

Figure 2: The overall architecture of the QPMRC-NER

vectors and $e_{i,1}, ..., e_{i,n}$ represent tokens of the $i$-th entity type name.

The model inputs are illustrated in Figure 2. We use the concatenation of context sentence and queries as the token inputs and assign the two parts with different segment ids, namely $u_a$ and $u_b$. We use the absolute position of tokens within each query or context sentence as the position ids, which guarantees the continuous prompt share the same position ids across queries. This precludes the occurrence of position confliction, as the queries are isolated from each other by virtue of the query-segmented attention mechanism proposed below.

**Query-Context Interaction via Query-Segmented Attention** The attention mechanism plays a critical role in modeling interactions between tokens in the transformer-based backbone. With self-attention, tokens in different queries could interact with each other. While modeling the dependencies between multiple entity types may be beneficial in some occasions, it adds difficulty for generalizing to new entity types in low resource setting.

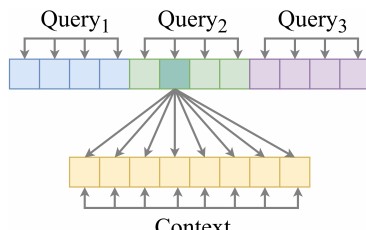

Figure 3: An illustration of Query-Segmented Attention

In order to address this issue, we propose the query-segmented attention, which is employed as a substitute for the self-attention component in the transformer architecture. The query-segmented attention segregates queries from one another and models the query-context interaction with a unidirectional flow as depicted in Figure 3. This mechanism enables context tokens to only attend to other context tokens, while query tokens can attend to tokens within the same query and to tokens in the context, allowing them to recognize the most pertinent parts of the context according to the entity type.

Thus, the query-segmented attention can be formulated as:

$$Q = H^{l-1}W_l^Q, \quad K = H^{l-1}W_l^K \qquad (1)$$

$$M_{ij} = \begin{cases} 0, & \text{allow to attend} \\ -\infty, & \text{prevent from attending} \end{cases} \qquad (2)$$

$$A_l = \text{softmax}\left(\frac{QK^\top}{\sqrt{d_k}} + M\right)(H^{l-1}W_l^V) \quad (3)$$

where parameters $W_l^Q, W_l^K, W_l^V \in \mathbb{R}^{d_h \times d_a}$ project the previous layer's output $H^{l-1} \in \mathbb{R}^{d_x \times d_h}$ to queries, keys and values respectively. The mask matrix $M \in \{0, -inf\}^{d_x \times d_x}$ controls whether two tokens can attend to each other by setting its values to 0 or $-inf$. $A_l \in \mathbb{R}^{d_x \times d_a}$ represents the output of the query-segmented attention.

After the transformer encoding, the outputs of the last-layer are used to acquire the query and

context representations. For each query $Q_i$, the representation of its first token in the prompt is employed as the query representation $h^{q_i}$. For each context word $w_j$, its representation $h_j^c$ is obtained by aggregating the corresponding word-piece token representations through mean pooling.

**Parameter-Efficient Tuning** The query-parallel encoder is initialized with the weights of a pre-trained BERT model (Devlin et al., 2019). Rather than fine-tuning the BERT encoder directly, we adopt a parameter-efficient tuning strategy that enables effective tuning of a pretrained language model by updating only a small number of extra parameters while keeping most of the pretrained parameters frozen, thereby making the tuning process more data-efficient (Li and Liang, 2021; Pan et al., 2022). Specifically, our approach employs UNIPELT (Mao et al., 2022) as the parameter-efficient tuning method, which integrates several existing parameter-efficient tuning techniques as sub-modules and controls them via gating mechanism. Further details are in Appendix C.

## 2.3 Query-Conditioned Biaffine Predictor

After obtaining the context and query representations from the query-parallel encoder, we feed them to an entity predictor to generate the entity predictions. The biaffine classifier (Yu et al., 2020) has been demonstrated to be an effective approach for entity prediction in prior studies, which jointly predicts the start and end positions of an entity span as well as the entity types. However, the prior biaffine approaches are employed in the span prediction paradigm without leveraging the the semantics of entity type names, which limits its ability to generalize to new entity types.

In this work, we propose a query-conditioned biaffine predictor for entity prediction, which integrates the biaffine predictor into the query-parallel machine reading comprehension framework and could utilizing the semantic meaning of entity type names. For each query $Q_i$, we first obtain two query-conditioned representations $h^{s_i}$ and $h^{e_i}$ corresponding to the start and end positions of entity spans through feed-forward neural networks:

$$h_j^{s_i} = \text{gelu}((h^{q_i} \oplus h_j^c)W_s + b_s) \quad (4)$$

$$h_j^{e_i} = \text{gelu}((h^{q_i} \oplus h_j^c)W_e + b_e) \quad (5)$$

where gelu is an activation function (Hendrycks and Gimpel, 2016), $W_s, W_e \in \mathbb{R}^{2d_h \times d_p}$, $b_s, b_e \in \mathbb{R}^{d_p}$ are learnable parameters.

Then we predict whether a span starting from position $j$ and ending at position $k$ belongs to certain entity type $i$ by:

$$r_{j,k}^i = h_j^{s_i \top} U_m h_j^{e_i} \\ + (h_j^{s_i} \oplus h_j^{e_i} \oplus l_{k-j})W_m + b_m \quad (6)$$

$$y_{j,k}^i = \text{sigmoid}(r_{j,k}^i) \quad (7)$$

where $l_{k-j} \in \mathbb{R}^{d_p}$ represents the $(k-j)$-th width embedding from a learnable look-up table, $U_m \in \mathbb{R}^{d_p \times 1 \times d_p}$, $W_m \in \mathbb{R}^{(2d_p+d_l) \times 1}$ and $b_m \in \mathbb{R}^1$ are learnable parameters. Different from traditional bi-affine predictor, the probabilities of an entity span belong to each entity type are predicted independently without inter-type competition, thus allowing for the sharing of predictor parameters across different entity types, which is beneficial for entity type expanding. To avoid conflict predictions in flat NER tasks during inference, we only keep the span with the highest prediction probability for any overlapping entity spans.

Besides the span-level prediction, we suggest a token-level auxiliary task to enhance the quality of the context and query representations. This task predicts whether a token is associated with a particular entity type through a feed-forward neural network as follows:

$$h_j^{t_i} = \text{gelu}((h^{q_i} \oplus h_j^c)W_t + b_t) \quad (8)$$

$$y_j^{i'} = \text{sigmoid}(h_j^{t_i}W_t' + b_t') \quad (9)$$

For both the span and token level tasks, we use the binary cross entropy function as the loss metric:

$$\mathcal{L}_s = - \sum_{0 \le j \le k \le n} \sum_{i \in \mathcal{E}} \mathbb{1}[\hat{y}_{j,k}^i = 1] \log y_{j,k}^i \\ + \mathbb{1}[\hat{y}_{j,k}^i = 0] \log(1 - y_{j,k}^i) \quad (10)$$

$$\mathcal{L}_t = - \sum_{0 \le j \le n} \sum_{i \in \mathcal{E}} \mathbb{1}[\hat{y}_j^i = 1] \log y_j^i \\ + \mathbb{1}[\hat{y}_j^i = 0] \log(1 - y_j^i) \quad (11)$$

where $\mathcal{L}_s$ and $\mathcal{L}_t$ are span level and token level losses, $\hat{y}_{j,k}^i$ and $\hat{y}_j^i$ are the corresponding labels. Then the final loss $\mathcal{L}$ on training set $D$ is:

$$\mathcal{L} = \sum_D \mathcal{L}_s + \lambda \mathcal{L}_t \quad (12)$$

where $\lambda$ is a coefficient.

## 3 Experiments

### 3.1 Setups

**Evaluation Tasks** To verify the effectiveness of the proposed method, we conduct extensive experiments in both resource-rich and resource-limited NER settings:

- Standard supervised NER setting: This task evaluates the efficacy of a model when adequate labeled data is available for training.

- Training-from-scratch low-resource NER setting: This task assess models' effectiveness under conditions of limited training data.

- In-domain low-resource NER setting: In this task, models are trained on a domain where some entity types have a sufficient number of labeled data while others have only a limited amount of labeled data, reflecting a common scenario where new entity types emerge in existing text domain. This could demonstrate the model's capacity for in-domain transfer.

- Cross-domain low-resource NER setting: This task involves training models on a resource-rich source domain, followed by fine-tuning on a target domain with different entity types and a limited amount of labeled data, demonstrating the model's capacity for cross-domain transfer.

**Datasets** For standard supervised NER setting, we conduct experiments on three datasets: CoNLL03 (Tjong Kim Sang and De Meulder, 2003), MSRA (Levow, 2006) and GENIA (Ohta et al., 2002), which are English flat NER dataset, Chinese flat NER dataset and English nested NER dataset respectively. For training-from-scratch low-resource setting, we use the MIT Restaurant Review (Liu et al., 2013), MIT Movie Review (Liu et al., 2013), and Airline Travel Information Systems (ATIS) (Hakkani-Tür et al., 2016) datasets. For the in-domain low-resource setting, we use the CoNLL03 (Tjong Kim Sang and De Meulder, 2003). For cross-domain low-resource setting, we use the CrossNER dataset (Liu et al., 2021) which contains five diverse domains, including politics, natural science, music, literature and artificial intelligence.

**Baselines** We conduct a comprehensive comparison of QPMRC-NER with several representative

| Method | English CoNLL03 | | |
| --- | --- | --- | --- |
| | Pre. | Rec. | F1 |
| BERT-Tagger | 91.93 | 91.54 | 91.73 |
| TemplateNER | 90.51 | 93.34 | 91.90 |
| PIQN | 93.29 | 92.46 | 92.87 |
| BERT-MRC† | 92.47 | 93.27 | 92.87 |
| LightNER | 92.39 | 93.48 | 92.93 |
| BS-NER | **93.61** | **93.68** | **93.65** |
| QPMRC-NER | 93.29 | 92.78 | 93.03 |
| Method | Chinese MSRA | | |
| | Pre. | Rec. | F1 |
| BARTNER | 93.21 | 91.97 | 92.58 |
| PIQN | 93.61 | 93.35 | 93.48 |
| BERT-MRC† | 96.18 | 95.12 | 95.75 |
| BS-NER | 96.37 | **96.15** | **96.26** |
| QPMRC-NER | **96.49** | 95.12 | 95.80 |
| Method | GENIA | | |
| | Pre. | Rec. | F1 |
| BERT-MRC† | 81.25 | 76.36 | 78.72 |
| BARTNER | 78.87 | 79.60 | 79.23 |
| Pyramid | 80.31 | 78.33 | 79.31 |
| PIQN | **83.24** | 80.35 | **81.77** |
| QPMRC-NER | 78.87 | **80.87** | 79.86 |

Table 1: Results for standard supervised NER. † means the result produced by (Yan et al., 2021)

methods as well as state-of-the-art approaches, including the classic BiLSTM-CRF (Ma and Hovy, 2016) and BERT-Tagger (Devlin et al., 2019), single query machine reading comprehension method BERT-MRC (Li et al., 2020), parameter-efficient tuning method LightNER (Chen et al., 2022), template-based low-resource NER method TemplateNER (Cui et al., 2021), cross-domain specialized method DoSEA (Tang et al., 2022), and several recently proposed supervised methods including Pyramid (Wang et al., 2020), BARTNER (Yan et al., 2021), PIQN (Shen et al., 2022), and Boundary Smoothing NER (BS-NER) (Zhu and Li, 2022). See Appendix B for a detailed introduction of those models.

**Implementation Details** We use pretrained BERT (Devlin et al., 2019) to initialize our encoder. For a fair comparison, we use BERT-large on CoNLL03, MIT Restaurant Review, MIT Movie Review and ATIS, BERT-base on CrossNER, BioBERT-large (Chiu et al., 2016) on GENIA and Chinese-BERT-WWM (Cui et al., 2020) on Chinese MSRA. We utilize the AdamW Optimizer (Loshchilov and Hutter, 2019) with a cosine annealed warm restart schedule (Loshchilov and Hutter, 2017) to train our model. In order to as-

| Method | MIT Movie | | | | | | MIT Restaurant | | | | | | ATIS | | |
|---|---|---|---|---|---|---|---|---|---|---|---|---|---|---|---|
| | 10 | 20 | 50 | 100 | 200 | 500 | 10 | 20 | 50 | 100 | 200 | 500 | 10 | 20 | 50 |
| BERT-Tagger | 25.2 | 42.2 | 49.6 | 50.7 | 59.3 | 74.4 | 21.8 | 39.4 | 52.7 | 53.5 | 57.4 | 61.3 | 44.1 | 76.7 | 90.7 |
| TemplateNER | 37.3 | 48.5 | 52.2 | 56.3 | 62.0 | 74.9 | 46.0 | 57.1 | 58.7 | 60.1 | 62.8 | 65.0 | 71.7 | 79.4 | 92.6 |
| BERT-MRC | 18.7 | 48.3 | 55.5 | 62.5 | 80.2 | 82.1 | 12.3 | 37.1 | 53.5 | 63.9 | 65.5 | 70.4 | 35.3 | 63.2 | 90.2 |
| LightNER | 41.7 | 57.8 | 73.1 | 78.0 | 80.6 | 84.8 | 48.5 | 58.0 | 62.0 | 70.8 | 75.5 | **80.2** | 76.3 | 85.3 | 92.8 |
| QPMRC-NER | **61.8** | **76.3** | **80.2** | **82.7** | **84.5** | **86.2** | **51.1** | **58.9** | **70.3** | **73.2** | **76.4** | 79.0 | **91.9** | **94.4** | **94.8** |

Table 2: Results for training-from-scratch low-resource NER

sess the model performance, we utilize span-based evaluation metrics, where an entity is only considered accurate if both the entity boundary and entity type are correct. The F1-score is used as the metric for evaluation. See Appendix A for more detailed hyper-parameter settings.

## 3.2 Standard Supervised NER

We first evaluate our method under the standard supervised NER settings on the CoNLL03, MSRA and GENIA datasets. For the CoNLL03, we follow (Lample et al., 2016; Yu et al., 2020; Yan et al., 2021) to train the model with the combined data from the training and development sets. A comparison with the state-of-the-art methods are listed in Table 1. It is evident that QPMRC-NER, which has been designed for low-resource NER, surpasses several recently proposed strong baseline models in rich-resource settings and performs comparable with the state-of-the-art methods, suggesting that QPMRC-NER is also suitable for NER tasks with abundant training data.

## 3.3 Training-from-scratch Low-resource NER

We analyze the efficacy of our method under the training-from-scratch low-resource NER settings where only $K$ samples of each entity type are available for training. Following (Cui et al., 2021), we employ the MIT Restaurant Review, MIT Movie Review and ATIS datasets for model evaluation by randomly sampling a fixed number of training instances per entity type ($K$=10, 20, 50, 100, 200, 500 for MIT Movie and MIT restaurant, and $K$=10, 20, 50 for ATIS).

The experimental results shown in Table 2 demonstrates that our method achieves significant performance boosts over baseline approaches and can more effectively utilize low-resource data. Specifically, when compared to the traditional machine reading comprehension based method BERT-MRC,

| Method | PER | ORG | LOC* | MISC* | Avg |
|---|---|---|---|---|---|
| BERT-Tagger | 75.71 | 77.59 | 60.72 | 60.39 | 69.62 |
| TemplateNER | 84.49 | 72.61 | 71.98 | 73.37 | 75.59 |
| BERT-MRC | 91.14 | 72.71 | 66.24 | 65.04 | 75.77 |
| QPMRC-NER | **95.65** | **85.52** | **84.10** | **74.17** | **86.68** |

Table 3: Results for in-domain low-resource NER. The Avg represents micro-average F1-score of all entity types and * represents entity types with a limited amount of labeled data.

our method surpasses it significantly especially in cases where $K$ is small. For instance, the F1 score of our model in a 10-shot setting is either higher or comparable to the F1 score of BERT-MRC in a 50-shot setting across all three datasets, suggesting that our method is more suitable for low-resource NER setting, particularly in the machine reading comprehension NER paradigm.

## 3.4 In-domain Low-resource NER

We conduct experiments for in-domain low-resource settings on the CoNLL03 dataset, where only a limited amount of labeled data is available for some entity types. Following (Cui et al., 2021), we downsample 4,001 training instances, including 3,763 "ORG", 2,496 "PER", 50 "LOC", and 50 "MISC".

The evaluation results are presented in Table 3. We observe significant performance boosts on resource-rich entity type "PER" and "ORG", as well as resource-limited entity type "LOC". The performance gain on the entity type "MISC" is relatively smaller, which may due to the fact that the entity type name "miscellaneous entity" is more semantic ambiguous and difficult to be represented by the query compared to other entity types. Our method achieves +10.91% F1-score improvement on average compared to baseline methods, which

| Method | Pol. | Sci. | Mus. | Lite. | AI. | Avg |
|---|---|---|---|---|---|---|
| BiLSTM-CRF | 53.89 | 49.12 | 43.65 | 41.87 | 43.18 | 46.34 |
| BERT-Tagger | 66.56 | 63.73 | 66.59 | 59.95 | 50.37 | 61.44 |
| TemplateNER | 65.41 | 62.93 | 64.67 | 64.55 | 57.64 | 63.04 |
| LightNER | 71.63 | 65.55 | 71.62 | 65.8 | 57.46 | 66.41 |
| BERT-MRC | 70.23 | 67.25 | 70.64 | 62.53 | 62.77 | 66.68 |
| DoSEA | 75.52 | 71.69 | 73.10 | 68.59 | 66.03 | 70.99 |
| QPMRC-NER | **77.06** | **76.90** | **78.34** | **69.40** | **66.33** | **73.61** |

Table 4: Results for cross-domain low-resource NER. The Avg represents macro-average F1-score of all domains.

| Method | Music (Source: None) | | | Music (Source: CoNLL03) | | |
|---|---|---|---|---|---|---|
| | Prec. | Rec. | F1 | Prec. | Rec. | F1 |
| QPMRC-NER | 81.50 | **70.40** | **75.55** | 83.58 | 73.72 | **78.34** |
| w/o continuous prompt | 82.16 | 66.52 | 73.51 | 82.52 | 71.18 | 76.43 |
| w/o qs-attention$_1$ | 78.45 | 61.70 | 69.08 | 83.21 | **73.83** | 78.24 |
| w/o qs-attention$_2$ | 81.11 | 61.70 | 70.09 | 80.25 | 61.59 | 69.69 |
| w/o query-parallel biaffine | 81.36 | 66.49 | 73.18 | 82.14 | 72.29 | 76.90 |
| w/o auxiliary task | **84.16** | 68.64 | 75.00 | **83.60** | 73.24 | 78.07 |

Table 5: Results of ablation studies. The experiment is conducted on the CrossNER music domain dataset with training-from-scratch and cross-domain low-resource settings. The "qs-attention$_1$" and "qs-attention$_2$" denote two variation models in query-segmented attention ablation study described in 3.7.

demonstrates it is more suitable for in-domain transfer and handling new emerging entities.

## 3.5 Cross-domain Low-resource NER

We explore the cross-domain low-resource NER settings in which models are initially trained in a resource-rich source domain and then fine-tuned and evaluated in a target domain with limited labeled data. In order to evaluate the models on different target domains with domain specific entity types, the CrossNER dataset is used as it is specifically designed for this task. It uses the CoNLL03 as the source domain and covers five distinct target domains, namely music, literature, artificial intelligence, politics, and natural science. There are 100 training instances for the first three domains and 200 instances for the last two domains. The empirical results in Table 4 indicate our approach surpasses existing state-of-the-art methods in all target domains, suggesting that our model is more suitable for cross-domain knowledge transfer.

## 3.6 Inference Speed

Table 6 presents the results of our method's inference speedup in comparison to BERT-MRC on the MIT Movie and ATIS datasets, demonstrating a

| Datasets | # Entity Type | BERT-MRC | QPMRC-NER |
|---|---|---|---|
| MIT Movie | 12 | 1.00× | 1.32× |
| ATIS | 79 | 1.00× | 2.16× |

Table 6: Results for inference speedup. Experiments are conducted on NVIDIA Tesla V100 Graphics Card with 32GB graphical memory.

respective speedup of 1.32× and 2.16×. The observed acceleration in inference speed is attributed to the query parallel setting employed by QPMRC-NER, which enables a single forward pass for all entities, as opposed to BERT-MRC's requirement of a separate forward pass for each entity. The degree of runtime improvement is contingent upon the number of entity types present in the dataset, with datasets featuring a greater number of entity types exhibiting a greater potential for leveraging the query parallel setting.

## 3.7 Ablation Studies

We conduct ablation studies to analyze the effects of different components of our model and validate the design decisions. Specifically, four settings are evaluated in the ablation studies: (1) **Shared**

**Prompt**: In our model, a query representing an entity type is composed by combining the shared continuous prompt with the entity type name. We ablate the shared continuous prompt to testify whether it is helpful for entity type representation and knowledge transfer across domains. (2) **Query-Segmented Attention**: The query-segmented attention used in the query-parallel encoder isolates the queries from each other and allows queries to attend to the context but not vice versa. We verify the effectiveness of query-segmented attention through two variations by relaxing the constraints. The first variation permits the context tokens to attend to the queries, thereby enabling the semantics of queries to interact indirectly through the context. The second one applies vanilla self-attention directly, and the first token of each entity type name is utilized to represent the queries. (3) **Query-Conditioned Biaffine Predictor**: To evaluate the effectiveness of the query-conditioned biaffine predictor utilized in our model, we conduct an experimental analysis by removing it and only employing word-level classification for named entity recognition. (4) **Auxiliary Task**: We access the effectiveness of the word level classification auxiliary task by removing it and only use $\mathcal{L}_s$ as training object.

This experiment is conducted in both training-from-scratch and cross-domain low-resource settings to assess the models' capacity to utilize small amounts of data and transfer knowledge across domains. The CrossNER music domain data is employed for evaluation. The experimental results, as presented in Table 5, demonstrate that the proposed model exhibits superior performance in the cross-domain settings as compared to the training-from-scratch setting, suggesting its ability to transfer knowledge from the source domain to the target domain, thereby enhancing its overall performance. Notably, QPMRC-NER outperforms its ablation variations in both settings. The experimental results indicate that shared continuous prompting is beneficial in improving query representation quality. Furthermore, the query-segmented attention mechanism is found to be more efficient than other attention mechanisms in our query-parallel MRC-based framework. This may be attributed to its ability to leverage the shared query prompt while avoiding modeling entity type interaction with low-resource data, which can have negative effects, particularly in the cross-domain setting. As evidenced by the experimental results, the query-segmented

attention mechanism outperforms the vanilla self-attention mechanism by a significant margin of 8.65% in the cross-domain setting. Additionally, the query-conditioned biaffine predictor, designed for the query parallel NER framework, achieves superior performance compared to using word-level classification directly. Also, incorporating word-level classification as an auxiliary task can further enhance the model's performance.

## 4 Related Work

Although NER is usually formalized as a sequence labeling task (Chiu and Nichols, 2016; Ma and Hovy, 2016; Xia et al., 2019; Devlin et al., 2019), several new NER paradigms have been proposed recently, conceptualizing NER as span classification (Luan et al., 2019; Yu et al., 2020; Li et al., 2021; Shen et al., 2021; Fu et al., 2021a), sequence generation (Straková et al., 2019; Yan et al., 2021; Tan et al., 2021; Lu et al., 2022), constituency parsing (Fu et al., 2021b; Lou et al., 2022) and machine reading comprehension (Li et al., 2020; Mengge et al., 2020) tasks and achieving impressive results. However, these approaches are mainly focus on standard supervised setting, which is not suitable for low-resource scenarios. Another line of research sought to address the low-resource NER task using techniques such as prototype-based learning (Fritzler et al., 2019; Yang and Katiyar, 2020; Henderson and Vulić, 2021), template-based learning (Cui et al., 2021) and contrastive learning (Das et al., 2022). But they often fail to fully exploit the potential of pretrained language models (PLMs) and perform inferior to standard sequence labeling NER methods in resource-rich settings. To bridge the gap, we propose a machine reading comprehension based method, which is effective in both resource-rich and resource-limited NER settings.

## 5 Conclusion

In this paper, we propose a query-parallel machine reading comprehension framework, which predicts all entities simultaneously and is applicable to both resource-rich and resource-limited settings. Specifically, we introduce the query-parallel encoder, which leverages the query-segmented attention mechanism to facilitate more straightforward generalization to new entity types. Additionally, we propose the query-conditioned biaffine predictor, which enables parallel prediction of entities. The

model is trained with parameter-efficient technique for data-efficiency. Extensive experimentation has shown that our approach attains a faster inference speed, exhibits competitive performance against strong baselines in resource-rich setting, and yields state-of-the-art outcomes in low-resource contexts, including training-from-scratch, in-domain transfer, and cross-domain transfer tasks.

## Limitations

In this work, we propose a query-parallel machine reading comprehension framework for NER task, which extracts multiple entity types simultaneously and achieve promising results for both resource-rich and resource-limited settings. In this approach, each query is semantically isolated, responsible for giving semantic signals to the pretrained language model and extracting entities of its type. The method is effective in low-resource settings when the entity type name is semantically unambiguous, but encounters difficulties otherwise, such as facing the miscellaneous other-class words. Thus, further research is needed to determine how to address these entity types and improve the performance of the model. Moreover, in QPMRC-NER, the input length restriction imposed by pretrained language models limits the number of parallel processed queries. To handle fine-grained entity extraction with a large number of entity types, segmentation of queries into multiple groups and separate encoding becomes necessary. Another potential approach is using a lightweight model to filter out irrelevant entity types for each sentence, thereby retaining only a small subset of entity type candidates. This avenue of investigation is left for future research.

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

## A  Hyper-parameters

The detailed hyper-parameters used in our model are listed in Table 7. For the parameter-efficient tuning module UNIPELT (Mao et al., 2022), we adopt the original hyper-parameter settings.

| Hyper-parameter | Value |
|---|---|
| **General** | |
| predictor hidden size | 300 |
| prefix length | 10 |
| adapter bottleneck size | 48 |
| LoRA rank | 8 |
| prompt length | 10 |
| learing rate | [8e-5, 1e-4, 1.2e-4] |
| batch size | 32 |
| auxiliary task coefficent | [0.1, 1] |
| warmup ratio | 0.01 |
| **Resource-limited Setting** | |
| max steps | [10000, 20000] |
| restart steps | [2000, 3000] |
| **Resource-rich Setting** | |
| epochs | 100 |
| restart epoch | 1 |

Table 7: Hyper-parameters for our model

## B  Baselines

We use the following methods as baselines:

**BiLSTM-CRF** (Ma and Hovy, 2016) treats NER as a sequence labeling task utilizing BiLSTM and CRF.

**BERT-Tagger** (Devlin et al., 2019) treats NER as a sequence labeling task utilizing BERT.

**BERT-MRC** (Li et al., 2020) formulates NER as a machine reading comprehension task. It extracts entities from the context based on manually-crafted queries representing different entity types.

**Pyramid** (Wang et al., 2020) a layered neural model for nested entity recognition, which permits each decoding layer to consider the global information from both the upper and lower layers.

**BARTNER** (Yan et al., 2021) formulates NER as an entity span generation task utilizing a sequence-to-sequence model with pointer mechanism.

**TemplateNER** (Cui et al., 2021) is a template-based method which treats NER as a language model ranking problem within a sequence-to-sequence framework and is designed for low-resource NER.

**LightNER** (Chen et al., 2022) formulates NER as an entity span sequence generation task. It adopts a sequence-to-sequence model with pointer and pluggable prompting mechanism to tackle low-resource NER task.

**PIQN** (Shen et al., 2022) treats NER as a span prediction problem by setting up learnable instance queries to extract entities from a sentence simultaneously.

**DoSEA** (Tang et al., 2022) treats NER as a machine reading comprehension based framework and is designed for cross-domain NER task. It is able to recognize distinctions that are domain-specific and mitigate the subtype conflicts between domains.

**Boundary Smoothing NER (BS-NER)** (Zhu and Li, 2022) treats NER as a span prediction problem and proposes the boundary smoothing regularization technique to boosting the model performance.

## C  Parameter-Efficient Tuning with UNIPELT

The UNIPELT (Mao et al., 2022) is utilized in our model as the parameter-efficient transfer learning component as illustrated in Figure 4, as it is able to adapt to the data or task setup dynamically. It incorporates three existing parameter-efficient tuning methods including Adapter (Houlsby et al., 2019), Prefix-tuning (Li and Liang, 2021) and LoRA (Guo et al., 2021; Hu et al., 2021) as sub-modules and controls them via gating mechanism.

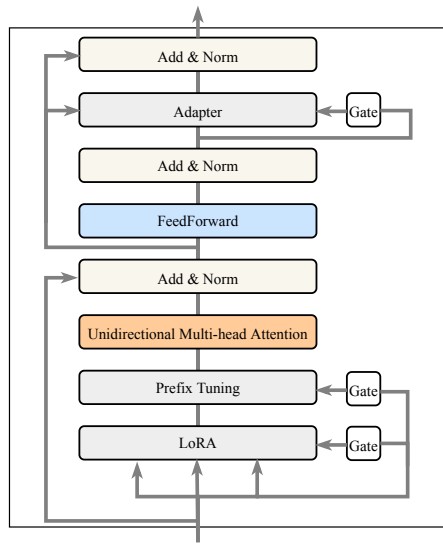

Figure 4: Illustration of the UNIPELT-enhanced transformer layer used in the query-parallel encoder

**Adapter** (Houlsby et al., 2019) augments the Transformer layer of the pretrained language model with a trainable bottleneck layer. This layer consists of a pair of down+up projections which reduce and then recover the size of the token hidden states. Mathematically, it could be denoted as:

$$H_A = W_{\text{up}}^\top \phi(W_{\text{down}}^\top H_F) \qquad (13)$$

**Prefix-tuning** (Li and Liang, 2021) enhances the multi-head self-attention in each Transformer layer by adding of a number of trainable vectors to the input, allowing the original tokens to attend to virtual tokens as if they are real.

**LoRA** (Guo et al., 2021; Hu et al., 2021) introduces trainable low-rank matrices and utilizes them to update the Query $Q$ and Value $V$ in multi-head self-attention in each Transformer layer, which can be formulated as:

$$Q = (W_Q^\top + \alpha_q W_{\text{up}}^{q\top} W_{\text{down}}^{q\top}) H_{\text{in}} \qquad (14)$$

$$V = (W_V^\top + \alpha_v W_{\text{up}}^{v\top} W_{\text{down}}^{v\top}) H_{\text{in}} \qquad (15)$$

**Gating Mechanism** The UNIPELT adds a trainable gate $\mathcal{G}_{m_i}$ for each sub-module $m_i \in \{Adapter, PrefixTuning, LoRA\}$ in the Transformer layer to achieve fine-grained control over these sub-modules. The gate output would be higher if its corresponding sub-module is more useful for the task.

Specifically, for adapter, we feed its direct input $H_F$ to a feedforward network to get the gating estimation $\mathcal{G}_A \in (0, 1)$. Then the output of adapter with gating mechanism $H_{A_\mathcal{G}}$ would be:

$$H_{A_\mathcal{G}} = \mathcal{G}_A H_A + H_F \qquad (16)$$

Similarly, for prefix-tuning, gating function $\mathcal{G}_P$ is applied to the prefix vectors $P_K$ and $P_V$. As for LoRA, the hyper-parameter $\alpha$ is substituted by the gating function $\mathcal{G}_L$.