# OpenReview forum: "A Query-Parallel Machine Reading Comprehension Framework for Low-resource NER"
_EMNLP/2023/Conference — EMNLP 2023 Findings_

### Official Review · Reviewer_HR2U · 2023-07-28

**Soundness:** 3

**Excitement:**

3: Ambivalent: It has merits (e.g., it reports state-of-the-art results, the idea is nice), but there are key weaknesses (e.g., it describes incremental work), and it can significantly benefit from another round of revision. However, I won't object to accepting it if my co-reviewers champion it.

**Missing References:**

It would be worth to also point to the distinct setting of few-shot NER (and clearly state the different model input/output in such works). A quick search immediately led me to this: https://paperswithcode.com/paper/a-comparative-study-of-pre-trained-encoders-1

**Paper Topic And Main Contributions:**

The paper considers NER, and particularly focuses on low-resource settings.
The authors frame it as a machine reading comprehension (MCR) task -- as do several recent works -- and proposes a model that jointly extracts multiple entity types at once. They thus propose feeding multiple queries (one for each entity type) to an encoder (BERT-based), propose an adapted attention mechanism (basically avoiding the multiple queries to attend to each other), use a biaffine classifier to which they also feed the entity type name, and add an auxiliary loss term on individual token classification (besides the span identification).
Experiments are presented in various settings: (1) large training data set, (2) low-resource training from scratch, (3) low-resource with transfer from a different large domain.
Results show competitive results for (1), and performance exceeding prior works for (2) and (3).
Contributions are mainly the new QPMRC-NER model with the above characteristics.

**Questions For The Authors:**

- Why did you not include DoSEA for all low-resource experiments (but only in Table 4)?
- Why does query 2 in Fig 2 have two entity names?
- What is the Type part (u_a and u_b tokens) of the input in Fig 2? You never explained this. Are these used to determine M_{i,j}?
- Are the chosen BERT encoders the same as those in the baseline models you compare against? (If not, could performance stem from just using a better BERT?)  [This kind of information should be clarified more explicitly in the paper]


**Reasons To Accept:**

- New multiple-entity-types-at-once NER system based on BERT and biaffine encoding; model dubbed QPMRC-NER (the authors may want to think about an easier mnemonic name though)
- Results showing the benefits compared to recent works in terms of better NER performance in low-resource settings, and speed (~ up to 2x)


**Reasons To Reject:**

- Model & experiment details do not suffice for independent reproduction of the model implementation nor experimental results
- Not all experiments seem to use the better competitor model (cf. only Table 4 includes DoSEA, where it seems to be the strongest competitor; hence why not include it in other low-resource oriented experiments?)
- Not specified which performance metrics are reported exactly (e.g., Table 2: is this micro- or macro-average F1, over all entity types? are the latter the same in all 3 datasets?)

**Reproducibility:**

3: Could reproduce the results with some difficulty. The settings of parameters are underspecified or subjectively determined; the training/evaluation data are not widely available.

**Reviewer Confidence:**

3: Pretty sure, but there's a chance I missed something. Although I have a good feel for this area in general, I did not carefully check the paper's details, e.g., the math, experimental design, or novelty.

**Typos Grammar Style And Presentation Improvements:**

General comments:
- please ensure all symbols are properly defined (e.g., d_x, d_h, d_a are what kind of dimensions?)
- please unambiguously state what performance metric is reported (e.g., Table 1 are micro- or macro-averages over the different entity types?)
- please indicate meaning of bold/underline values in tables. (Btw, please always indicate the best results; why don't you do it in Table 1?)
- please do not use vertical lines in tables (e.g., personally, I found this quite helpful to create nice looking tables: https://people.inf.ethz.ch/markusp/teaching/guides/guide-tables.pdf)

Details:
- 026: predictor to predict entities simultaneously -> predictor to extract multiple entities at once
- 029: efficiency -> efficient
- 030: perform -> performs
- 032: setting -> settings
- 074: employing the -> employing
- 075: The prior -> Prior
- 081: demonstrated the ability to achieve ... -> demonstrated ...
- 097: "continuous prompt": not clear what this means. How can a prompt be "continuous"? I'd expect it to be "static", or possibly "dynamic" if it's template-based...; it would be highly useful to include a concrete example (e.g., in Fig 2 use an actual prompt rather than just abstract identifiers  p_0 ... p_m)
- 107: "parameter-efficient techniques for data efficiency" -> overly vague; if you don't want to give more details here, just say it is "data efficient" and point to a reference / later part of the paper where you detail what you're talking about
- 116: facilitates the -> facilitates
- 118: thereby resulting -> resulting
- 127-134: overly verbose; I suggest:
    Section 2.1 presents the task formulation and introduces our QPMRC-NER method, illustrated in Fig. 2. QPMRC-NER comprises (i) a query-parallel encoder and (ii) a query-conditioned biaffine predictor, detailed in §2.2 and §2.3 respectively.
- 142: "a query Q for each entity type": it seems this query is static and always the same for all entity types (cf. p_0...p_m remains the same for Q1, Q2... in Fig.2)? Please be clear about this. Also, are p_i actual words/tokens, or arbitrary learnable embeddings?
- 150: "context": is this the actual input sentence to perform NER on? Or does it contain something else/more? Please be clear
- 153-54: that can be further used -> that are used
- Fig 2:
  + please clearly define all symbols
  + clarify the meaning of the matrices in the top right (I assume they represent y^i_{k,l}, scoring span (k,l) with probability of representing an entity of type i ?
  + Encoder: why not be more specific, and say "BERT-based encoder"?
  + Not clear whether you train independent Biaffine Predictors for each type, or whether they all share the same parameters
  + why does Q2 have 2 entity type tokens e_{2,1} and e_{2,2}? seems arbitrary
- 170: is -> are
- 184: "transformer-based": please also already state which specific model(s) you're using in the experiments
- 202-208: not clear whether you define separate attention parameters W for each entity type, or if these are shared
- 209-210: please define the d_* parameters
- 212, 214: inf -> <infinity symbol>
- 217: utilized -> used
- 222: token -> word-piece token
- 238: For further details, please refer to Appendix C -> Further details are in Appendix C
- 287: overlapped -> overlapping
- 291-92: token is  ... type -> token $i'$ is .... type $j$
- 296: explain what y^{i'}_j represents  (see also 291-92)
- 331: distinct -> different
- 366: classic -> the classic
- Table 1:
    + please highlight best/runner-up results, as you do in Table 2
    + caption: reproduced -> produced
- 368: for detailed -> for a detailed
- Table 3:
   + I'd suggest to include a top row with the number of instances per entity type (which now is listed in the paper text instead)
   + what does the * indicate?
   + last row = macro-average over previous columns? or is it micro-average over all instances?
   + Would it make sense to repeat the low-resource sampling of just 50 LOC/MISC instances multiple times, and then report averages over these runs, to have a more robust result (with some confidence interval instead of just a single value)?
- 427: under -> for
- 428: setting -> settings
- Table 4:
   + are these micro/macro averages of F1 over all instances / entity types?
   + do all domains have the exact same entity types? in the same proportions? (If not, then maybe differences across them may simply stem from their different entity type distributions, not because of the domain per se)
- §3.6: please indicate hardware information
- 483: avoid line break after item number '(1)'  (hint: in LaTeX, use the enumitem package with the 'inline' option)
- §3.7:
   + I don't understand how exactly the uni-attention ablations work; please expand your explanation
   + same issue with the "query-conditioned biaffine predictor"; does it mean you now only use loss L_t from eq. (12) and thus essentially do individual token classification rather than span identification?
- 524: indicating -> indicates
- 541: However -> Also    [doesn't sound like a contrasting statement, rather something new/extra]
- 535: "vanilla self-attention": is this the same as your "w/o uni-attention_2"? Please be consistent in naming your ablation models
- §4.1
   + if it's the only subsection within §4, why name it?
   + seems to be largely a repetition + slight expansion of what you already stated in §1; please avoid duplication of similar statements
- 468: "inferior": compared to what alternative?
- 574: could predict -> predicts    [hint: https://learnenglish.britishcouncil.org/grammar/english-grammar-reference/can-could]
- 594: could extract -> extract
- 597: semantic -> semantically
- 600: "It": not 100% clear what this refers to
- 602: could encounter -> seems to encounter  [or just plainly: 'encounters']
- 608-9: "query isolation restriction": not clear what you mean exactly

---

> ### Author Rebuttal · Authors · 2023-08-29
>
> We sincerely thank the reviewer for the detailed feedback and invaluable suggestions.
>
> Q: Not all experiments seem to use the better competitor model. (cf. only Table 4 includes DoSEA, where it seems to be the strongest competitor; hence why not include it in other low-resource oriented experiments?)  Why did you not include DoSEA for all low-resource experiments (but only in Table 4)?
>
> A: DoSEA (DoSEA: A Domain-specific Entity-aware Framework for Cross-Domain Named Entity Recogition, Coling 2022) is a NER method designed specifically for cross-domain NER, which can identify domain-specific semantic differences and mitigate the subtype conflicts between domains and achieve sota performance as illustrated in Appendix B. Since its designed and only suitable for cross-domain NER, we only compare QPMRC-NER with it in the cross-domain low-resource NER setting.
> Low-resource NER encompasses various distinct settings, each characterized by its unique attributes. Certain NER methods are exclusively tailored to address the requirements of a specific setting, while others possess a versatility that allows them to be applicable across multiple settings. In our experimental investigation, we adopt a comprehensive approach by incorporating representative methods alongside state-of-the-art approaches as baseline methods. This selection encompasses models specifically devised to handle diverse low-resource settings, such as BERT-MRC and LightNER, as well as task-specific methodologies like DoSEA.
> ___
>
> Q: Not specified which performance metrics are reported exactly (e.g., Table 2: is this micro- or macro-average F1, over all entity types? are the latter the same in all 3 datasets?)
>
> A: The metric chosen to assess the performance of all experiments is the micro-average F1-score. We will provide a more explicit elucidation of this choice in our final version.
> ___
>
> Q: Are the chosen BERT encoders the same as those in the baseline models you compare against? (If not, could performance stem from just using a better BERT?)
>
> A: The BERT encoders we use are the same as those in the baseline models in all experiments , as elucidated in the Implementation Detail section of Section 3.1.
> To elaborate, we utilize BERT-large for CoNLL03, MIT Restaurant Review, MIT Movie Review, and ATIS datasets. For the CrossNER dataset, we employ BERT-base, while for the GENIA dataset, we utilize BioBERT-large. Additionally, the Chinese MSRA dataset is processed using Chinese-BERT-WWM. We will make it more clearly in our final version.
> ___
>
> Q: Why does query 2 in Fig 2 have two entity names?
>
> A: The e_{i,j} represents the jth token of the entity i. In Figure 2, the query 2 serves as a representation of an entity type that encompasses a name consisting of two tokens. In our final version, we will provide a more explicit clarification to ensure better understanding and clarity.
> ___
>
> Q: What is the Type part (u_a and u_b tokens) of the input in Fig 2? You never explained this. Are these used to determine M_{i,j}?
>
> A: The u_a and u_b represent the segment ids for input sentence and queries respectively, enabling the model to differentiate between different input components. The details regarding the model inputs are expounded upon in the "Query Generation and Model Input" section of Section 2.2. We will explain the symbols and make it more clarity in our final version.
> ___
>
> Q: It would be worth to also point to the distinct setting of few-shot NER (and clearly state the different model input/output in such works). A quick search immediately led me to this: https://paperswithcode.com/paper/a-comparative-study-of-pre-trained-encoders-1
>
> A: Thank you for the suggestion. We will incorporate a relevant low-resource NER survey as a point of reference for a better distinction and understanding of the different NER setups encountered in low-resource contexts.
> ___
>
> Typos Grammar Style And Presentation Improvements
>
> We express our sincere gratitude for the comprehensive suggestions provided to enhance the presentation of our work. We will diligently address any typographical errors and refine the representations accordingly, taking into account the invaluable insights provided. Your suggestions will significantly contribute to improving the overall quality and clarity of our work.

---

### Official Review · Reviewer_Km8b · 2023-07-31

**Soundness:** 4

**Excitement:**

4: Strong: This paper deepens the understanding of some phenomenon or lowers the barriers to an existing research direction.

**Missing References:**

N/A.

**Paper Topic And Main Contributions:**

This paper introduces QPMRC-NER, a modified machine-reaching comprehension (MRC) based method for NER. The primary contribution is to extend the MRC-NER framework such that multiple queries can be issued in parallel, therefore improving efficiency during inference.

**Questions For The Authors:**

- On lines 106-107, it says, "The model is trained with parameter-efficient techniques for data-efficiency." Parameter-efficient techniques are used to reduce memory requirements, no? I do not understand what "data-efficiency" is referring to here.


**Reasons To Accept:**

- The MRC-based framework for NER is a popular approach. Research on this approach will greatly benefit from the query-parallel modifications proposed in this paper.
- The authors are careful to benchmark on many datasets, from different languages and domains, in resource-rich and resource-poor settings. Their model performs comparably or better in almost all cases, significantly outperforming previous approaches in the low-resource setting.

**Reasons To Reject:**

- It is a little unclear to me how this would scale to a setting of fine-grained entity extraction (https://ojs.aaai.org/index.php/AAAI/article/view/8122). At the very least, it would be helpful if the authors added a comment to the limitations section about how this would scale to cases where the number of entity types is large (e.g. 10s or even 100s).

**Reproducibility:**

4: Could mostly reproduce the results, but there may be some variation because of sample variance or minor variations in their interpretation of the protocol or method.

**Reviewer Confidence:**

3: Pretty sure, but there's a chance I missed something. Although I have a good feel for this area in general, I did not carefully check the paper's details, e.g., the math, experimental design, or novelty.

**Typos Grammar Style And Presentation Improvements:**

- I don't understand what the "Transfer" arrow in Figure 1 is trying to show. It almost appears to point from "Context" to "ORG/query". Could the authors make it more explicit when/how the transfer happens? Maybe updating the figure caption to explain.
- "Unidirectional attention" seems like a poor name, as it suggests a mechanism like the causal or masked self-attention of transformer decoders in language modelling. I also don't clearly understand how the modification in this paper is "unidirectional". Something like "query-segmented attention" would be more immediately understandable (IMO) and avoids the erroneous connection to causal attention.

---

> ### Author Rebuttal · Authors · 2023-08-29
>
> Thank you for dedicating your time to reviewing our paper and offering us invaluable feedback.
>
> Q: It is a little unclear to me how this would scale to a setting of fine-grained entity extraction (https://ojs.aaai.org/index.php/AAAI/article/view/8122). At the very least, it would be helpful if the authors added a comment to the limitations section about how this would scale to cases where the number of entity types is large (e.g. 10s or even 100s).
>
> A: Thank you for providing valuable suggestions. In the QPMRC-NER approach, the context sentence and queries representing different entity types are concatenated and fed into a BERT-like encoder. Consequently, the simultaneous prediction of entity types is constrained by the maximum input length allowed by the pretrained language model. However, our method is capable of accommodating a larger number of entity types (e.g., the ATIS dataset used in our study consists of 79 entity types) by utilizing shorter continuous prompts. In cases where the entity types are even larger, we can segment them into multiple parts and feed them into the encoder separately. Nevertheless, additional post-processing steps are required to prevent conflicting predictions in the flat NER task. Alternatively, employing a lightweight model to pre-filter irrelevant entity types for each sentence and retaining only a small subset of entity type candidates could be a considerable approach. However, this avenue of investigation is left for future research. We will provide a more comprehensive discussion on the scalability of QPMRC-NER for fine-grained entity extraction in the final version of our paper.
> ___
>
> Q: On lines 106-107, it says, "The model is trained with parameter-efficient techniques for data-efficiency." Parameter-efficient techniques are used to reduce memory requirements, no? I do not understand what "data-efficiency" is referring to here.
>
> A: In the context of our study, data efficiency pertains to a model's capacity to achieve a satisfactory level of generalization performance with a limited quantity of data. Employing parameter-efficient techniques, as opposed to directly fine-tuning the model, can mitigate memory requirements while simultaneously enhancing the effectiveness of data utilization.
> ___
>
> Q: I don't understand what the "Transfer" arrow in Figure 1 is trying to show. It almost appears to point from "Context" to "ORG/query". Could the authors make it more explicit when/how the transfer happens? Maybe updating the figure caption to explain.
>
> A: In Figure 1, the "Transfer" arrow is intended to depict the potential advantages that the proposed QPMRC-NER approach can obtain through the transfer of knowledge from a resource-rich domain to a resource-limited domain. This transfer include in-domain and cross-domain transfer scenarios that are discussed in Section 3.4 and 3.5 in the paper. In our final version, we will enhance the clarity of the figure to better convey this concept.
> ___
>
> Q: "Unidirectional attention" seems like a poor name, as it suggests a mechanism like the causal or masked self-attention of transformer decoders in language modeling. I also don't clearly understand how the modification in this paper is "unidirectional". Something like "query-segmented attention" would be more immediately understandable (IMO) and avoids the erroneous connection to causal attention.
>
> A: Thank you for the suggestion. We will revise the term "Unidirectional attention" to a more immediately understandable name.

---

### Official Review · Reviewer_Nr7p · 2023-08-02

**Soundness:** 3

**Excitement:**

3: Ambivalent: It has merits (e.g., it reports state-of-the-art results, the idea is nice), but there are key weaknesses (e.g., it describes incremental work), and it can significantly benefit from another round of revision. However, I won't object to accepting it if my co-reviewers champion it.

**Paper Topic And Main Contributions:**

The paper proposed a query-parallel approach to enhance the MRC-based NER methods, which can extract multiple entity types concurrently and is applicable to both resource-rich and low-resource settings. The query-parallel encoder used a unidirectional flow to encode the query and context. And a query-conditioned biaffine predictor to predict entities of all types simultaneously. The authors conducted extensive experiments in resource-rich setting and low-resource settings and achieved superior perfornance.

**Reasons To Accept:**

1. The approach proposed by the paper is rather novel to me. This authors proposed entity type-specific query-parallel network on top of MRC-based NER method and PIQN method. It avoids both the multiple rounds of querying in MRC-based methods and the dynamic loss designment during training of PIQN network.
2. The experimental setup is sufficient and a variety of settings are considered. Although this method did not achieve the best results on some datasets, it also proved to have promising performance in different settings.

**Reasons To Reject:**

1. Some recent works need to be discussed and more works need to be compared in experiments. e.g.  PromptNER, COPNER, LightNER
2. The Inference Speed comparison is inadequate, and I would like to see more methods such as COPNER and PIQN.
3. More comparisons are needed to elucidate the difference and advantages with PIQN.

+ COPNER: Contrastive Learning with Prompt Guiding for Few-shot Named Entity Recognition (Huang et al., COLING 2022)
+ LightNER: A Lightweight Tuning Paradigm for Low-resource NER via Pluggable Prompting (Chen et al., COLING 2022)
+ PromptNER: Prompt Locating and Typing for Named Entity Recognition (Shen et al., ACL 2023)

**Reproducibility:**

3: Could reproduce the results with some difficulty. The settings of parameters are underspecified or subjectively determined; the training/evaluation data are not widely available.

**Reviewer Confidence:**

4: Quite sure. I tried to check the important points carefully. It's unlikely, though conceivable, that I missed something that should affect my ratings.

---

> ### Author Rebuttal · Authors · 2023-08-29
>
> Thank you for dedicating your time to reviewing our paper and offering us invaluable feedback.
>
> Q: Some recent works need to be discussed and more works need to be compared in experiments. e.g. PromptNER, COPNER, LightNER
>
> A: Thanks very much for bring to our attention some recent works.
>
> LightNER: LightNER is a state-of-the-art method for low-resource named entity recognition. We have included LightNER as one of our baseline methods, as demonstrated in the Baselines section of our paper (Section 3.1). In order to evaluate the efficacy of our method, we compare it against LightNER in both resource-rich and low-resource settings, and our method outperforms LightNER in both scenarios.
>
> PromptNER: PromptNER is an NER method that integrates entity locating and entity typing through prompt learning. We did not incorporate PromptNER as a baseline in our study because it was published in ACL 2023, which is subsequent to our submission to EMNLP 2023.
> There are notable distinctions between our method and PromptNER. In PromptNER, each entity is predicted using a single prompt, necessitating the estimation of the maximum number of entities per sentence in advance. Moreover, PromptNER employs a complex dynamic template filling mechanism to establish the correspondence between prompts and entities. Conversely, our approach employs a distinct methodology where each prompt is allocated for the prediction of entities associated with a specific type, thus mitigating the complexities mentioned earlier.
> Fortunately, we share the same experimental settings as PromptNER for training-from-scratch low-resource NER and in-domain low-resource NER, following a previous work TemplateNER (Cui et al. 2021). The comparative results are shown below.
>
> Results for training-from-scratch low-resource NER:
> | Method     | MIT Movie 10 | MIT Movie 20 | MIT Movie 50 | MIT Movie 100 | MIT Movie 200 | MIT Movie 500 | MIT Restaurant 10 | MIT Restaurant 20 | MIT Restaurant 50 | MIT Restaurant 100 | MIT Restaurant 200 | MIT Restaurant 500 | ATIS 10 | ATIS 20 | ATIS 50 |
> |--------------|--------------|--------------|--------------|----------------|----------------|----------------|--------------------|--------------------|--------------------|---------------------|---------------------|---------------------|---------|---------|---------|
> | PromptNER  | 55.6         | 68.2         | 76.5         | 80.4           | 82.9           | 84.5           | **56.1**           | **62.6**           | 69.3               | 71.3                | 74.4                | 77.4                | 91.5    | 94.3    | **95.5**    |
> | QPMRC-NER  | **61.8**     | **76.3**     | **80.2**     | **82.7**       | **84.5**       | **86.2**       | 51.1               | 58.9               | **70.3**           | **73.2**            | **76.4**            | **79.0**                | **91.9** | **94.4** | 94.8    |
>
> Results for in-domain low-resource NER:
>
> | Method     | PER   | ORG   | LOC*  | MISC* | Avg.   |
> |------------|-------|-------|-------|-------|--------|
> | PromptNER  | 76.96 | **88.11** | 82.69 | 62.89 | 79.75  |
> | QPMRC-NER | **95.65** | 85.52 | **84.10** | **74.17** | **86.68** |
>
> Additionally, our compare our method against PromptNER on the supervised NER setting using the CoNLL03 dataset with BERT-large encoder. The comparative results are presented as follows:
> | Method |  Precision | Recall | F1 |
> |---------|---------------|---------------|---------------|
> | PromptNER  | 92.48                    | 92.33                 | 92.41             |
> | QPMRC-NER  | **93.29**                | **92.78**             | **93.03**         |
>
> We can see that in general, our QPMRC-NER method performs either higher or comparable to PromptNER.
>
> COPNER: COPNER is an approach for few-shot NER that utilizes distance metric learning. It employs class-specific words as metric referents to infer the entity types of test tokens by comparing them with these referents. Our proposed method, QPMRC-NER, distinguishes itself from COPNER in several ways. QPMRC-NER adopts a machine reading comprehension-based approach to accurately predict entity spans directly. In contrast, COPNER relies on a metric learning-based approach that predicts the entity type of each token individually, without explicitly verifying entity boundaries. Consequently, COPNER necessitates the inclusion of the Viterbi decoding approach proposed in Structshot (Yang and Katiyar, 2020) to enhance the accuracy of its predictions.
> In our study, we adopt experimental setups similar to those employed in TemplateNER (Cui et al., 2021) and LightNER (Chen et al., 2022). These setups differ from those utilized in COPNER.
> Given that the training-from-scratch low-resource Named Entity Recognition (NER) setting is largely similar, we proceed to conduct experimental comparisons on the MIT-Movie dataset. Through these comparisons, our method exhibits superior performance in comparison to COPNER.
> | Method     | MIT Movie 10 | MIT Movie 20 | MIT Movie 50 | MIT Movie 100 | MIT Movie 200 | MIT Movie 500 |
> |------------|--------------|--------------|--------------|----------------|----------------|----------------|
> | COPNER  | 59.9         | 67.2         | 74.1         | 77.8           | 80.1          | 81.7           |
> | QPMRC-NER  | **61.8**     | **76.3**     | **80.2**     | **82.7**       | **84.5**       | **86.2**       |
>
>
> ___
>
> Q: More comparisons are needed to elucidate the difference and advantages with PIQN.
>
> A: PIQN is a Named Entity Recognition (NER) method that utilizes global and learnable instance queries to extract entities from sentences in a parallel manner. In contrast, our proposed QPMRC-NER differs from PIQN in terms of both algorithm design and application tasks.
> In PIQN, each entity is predicted by a specific prompt, similar to the approach employed in PromptNER. Consequently, it involves estimating the maximum number of entities per sentence and undergoing a dynamic label assignment process. In contrast, QPMRC-NER adopts a different approach by assigning each prompt to represent entities associated with a specific type, thereby avoiding such issues altogether.
> More importantly, it is worth noting that PIQN is primarily designed for supervised NER settings and not be suitable for low-resource NER tasks. We conducted tests on PIQN within various low-resource scenarios and found that its performance did not match that of other low-resource NER methods, such as BERT-MRC and LightNER. Consequently, we did not include PIQN as a baseline method for low-resource NER tasks. Instead, we employed representative and state-of-the-art methods from previous research for comparison purposes.
> Since QPMRC-NER can also be applied to standard NER settings, we included PIQN as one of the supervised NER baseline methods in our study. As indicated in Table 1, our method achieves superior or comparable performance compared to PIQN in this context.
> ___
>
> Q: Include more methods for inference speed comparison
>
> A: The proposed QPMRC-NER is a machine reading comprehension (MRC) based method that aims to enhance performance in both resource-rich and resource-limited NER tasks, while also improving inference speed compared to conventional MRC-based methods such as BERT-MRC.
> Given that different paradigms of NER methods like sequence labeling, machine reading comprehension, distant metric learning, and sequence generation can significantly impact inference speed, our primary comparative analysis centers around methods operating within the machine reading comprehension paradigm, specifically focusing on BERT-MRC. Notably, our proposed method demonstrates a significant speed improvement, achieving up to a 2x speedup when compared to BERT-MRC.

---

### Meta-Review · Area_Chair_1Jb1 · 2023-09-20

**Recommendation:** 2

**Metareview:**

This paper aims to improve MRC-style NER by introducing a query-parallel design, which can extract multiple entity types at the same time and works well for both resource-rich and resource-limited settings. Comprehensive evaluation on multiple datasets under different settings demonstrates the effectiveness of the proposed method.

Strengths:
- This is a solid improvement for MRC-style NER
- Comprehensive evaluation that demonstrates the effectiveness of the method especially under resource-limited settings

Weaknesses:
- As reviewer HR2U pointed out, there are many minor writing issues in the current paper that collectively makes it harder to read and follow
- There are concerns over reproducibility, especially for a complex method like this. There is no indication of code release, either in the paper or in the rebuttal when the reviewer pointed this out as a weakness.

---

### Decision · Program_Chairs · 2023-10-07

**Decision:**

Accept-Findings

**Comment:**

This paper aims to improve MRC-style NER by introducing a query-parallel design, which can extract multiple entity types at the same time and works well for both resource-rich and resource-limited settings. Comprehensive evaluation on multiple datasets under different settings demonstrates the effectiveness of the proposed method.

Strengths:
- This is a solid improvement for MRC-style NER
- Comprehensive evaluation that demonstrates the effectiveness of the method especially under resource-limited settings

Weaknesses:
- As reviewer HR2U pointed out, there are many minor writing issues in the current paper that collectively makes it harder to read and follow
- There are concerns over reproducibility, especially for a complex method like this. There is no indication of code release, either in the paper or in the rebuttal when the reviewer pointed this out as a weakness.